# Real-Time FTIR-ATR Spectroscopy for Monitoring Ethanolysis: Spectral Evaluation, Regression Modelling, and Molecular Insight

**DOI:** 10.3390/ijms26199381

**Published:** 2025-09-25

**Authors:** Jakub Husar, Lubomir Sanek, Jiri Pecha

**Affiliations:** Faculty of Applied Informatics, Tomas Bata University in Zlin, Nad Stranemi 4511, 760 05 Zlin, Czech Republic; sanek@utb.cz (L.S.); pecha@utb.cz (J.P.)

**Keywords:** FTIR-ATR spectroscopy, ethanolysis, vegetable oils, fatty acid ethyl esters, biofuels, online monitoring, real-time monitoring

## Abstract

As the demand for biodiesel continues to rise, there is a pressing need for efficient and continuous monitoring of the transesterification reaction at the industrial level. However, there is a lack of straightforward online monitoring methods capable of accurately following the course of ethanolysis under various reaction conditions. In this work, simple linear regression (SLR) and multiple linear regression (MLR) models were developed to assess Fourier transform infrared spectroscopy (FTIR) data from a continuous flow cell, enabling real-time ethanolysis monitoring without sample pretreatment. Gas chromatography (GC) was utilised as the reference method to accurately characterise the reaction mixture’s composition during ethanolysis. Extensive correlation analysis was performed to identify spectra regions where the reaction system’s state changes are observable. The gained regions were subsequently applied in the linear regression model’s development. This novel approach resulted in the performance of simple linear regression comparable to complex partial least squares (PLS) regression model (RMSEP = 2.11). The developed online monitoring system was validated in a wide range of reaction conditions (40–60 °C; 0.25–1.0% *w*/*w* NaOH); it effectively identifies dynamic changes in the ethanolysis process and confirms achieving the threshold value of ester content set by EU regulation directly in the production process.

## 1. Introduction

With the rapid development of industrial processes and the global surge in energy demand, the need for sustainable alternatives with a reliable process monitoring system becomes ever more pressing. The European Union, recognising this urgency, has set ambitious targets of achieving a minimum of 14% renewable energy in the transportation sector by 2030, including 3.5% from advanced biofuels. Biodiesel, emerging as a promising contender among alternative energy sources, plays a significant role in this transition [1,2]. Its roots trace back to Rudolf Diesel’s pioneering experiments with peanut oil as a substitute for diesel engines in the last century [3]. This renewable and biodegradable fuel, a blend of long-chain fatty acid alkyl esters, has found applications beyond biofuels, impacting diverse industries. From enhancing the effectiveness of agricultural sprays to serving as essential ingredients in drug formulations, fatty acid alkyl esters, the building blocks of biodiesel, prove to be versatile oleochemicals [4,5,6,7]. Their synthesis is predominantly achieved through a method known as transesterification. The transesterification reaction converts vegetable oils and animal fats (i.e., triglycerides), via successive reactions, into biodiesel, accompanied by the production of glycerol as a by-product [8]. While methanol is a commonly employed transesterification agent, ethanol is a promising alternative, offering environmental benefits [9,10]. Transesterification is prone to saponification [11]—an unwanted side reaction; consequently, ensuring optimal conditions, including nearly anhydrous environment, is crucial for achieving high conversion, yield and smooth downstream processing [12,13]. Hence, effective process control of biodiesel synthesis at the industrial level is essential for achieving and maintaining the desired product quality and for dynamic optimisation of the process efficiency [14]. Nevertheless, without a reliable feedback mechanism, managing process control becomes a challenging endeavour. Therefore, continuous straightforward and precise monitoring of the actual biodiesel content during production is imperative.

The quality of biodiesel, a fundamental factor in assessing overall production efficiency, is outlined in the standards [15,16,17]. To address this, various analytical methods have been designed, predominantly for offline monitoring at the end of the production process [18,19]. However, this approach is unsuitable for real-time (online) process monitoring. Despite the prevalent industrial standard of post-production quality control through offline analyses, these methods, while unsuitable for online control, provide valuable principles and techniques for designing online sensors and measurement systems. Gas chromatography (GC) is widely recognised as the predominant method for biodiesel composition analysis, endorsed by international standards [15,16,17,18]. It offers comprehensive quality assessment, including determination of alkyl esters, glycerides, and free glycerol contents [20]. High-performance liquid chromatography (HPLC), requiring no sample derivatisation, is another method with diverse detection approaches [21]. Spectroscopic methods such as 1H-Nuclear magnetic resonance (NMR) and 13C-NMR have been utilised for offline and real-time monitoring [22,23]. Near-infrared spectroscopy (NIR) and fluorescence spectroscopy provided online monitoring options with lower sensitivity than GC [24,25,26,27,28]. The viscosity difference between triglycerides and methyl esters was explored, with viscosity measurements aligning well with GC analysis [29,30]. Density deviation, impedance, speed of sound, and refractive index (RI) were investigated as cost-effective monitoring methods, with RI measurements showing promise for process control [30,31,32]. These varied analytical methods contribute to a comprehensive toolbox for monitoring biodiesel production, each with advantages and considerations. However, most of these methods involve offline procedures or lack the required accuracy for real-time monitoring, presenting challenges for seamless integration into supervisory control systems. While offline methods are generally applicable in process control, it is crucial to anticipate a significant time delay in the feedback loop, with an expected delay of 45 min (sample preparation and analysis), as illustrated in [20].

Spectroscopic data, providing molecular-level information, holds great promise for unravelling intricate mechanisms within complex reactive systems. Leveraging data-driven methods becomes crucial for implementing automated supervisory control, which ensures process safety and product quality. Methods involving Fourier transform infrared spectroscopy (FTIR) have been developed to track the course of transesterification. Yet, most are associated with offline procedures, including offline sample purification through centrifugation [33,34,35,36,37,38,39]. Available studies focused predominantly on methanolysis [33,34,35,36,37,38,40,41,42,43], with limited exploration of ethanolysis [39,44,45]. Calibration standards were typically formulated using blends, and validations employed various reference techniques, including GC, HPLC, NMR, etc. Spectra were obtained in a broad range, and authors explored specific regions or peak intensities without correction or applied baseline correction, smoothing, or other techniques. Different accessories, such as discs or windows with various crystals, have been utilised. While Attenuated total reflection (ATR)-FTIR is recognised for creating multivariate calibration models, it often employs only offline monitoring due to the need for sample purification [46]. However, some authors have explored online monitoring using probes or flow cells, providing representative spectral data for detecting changes in transesterification processes [40,42,44]. IR spectroscopy employing in situ ATR-FTIR has also proven valuable for studying of reaction mechanism of transesterification under supercritical conditions, revealing unique molecular interactions and homogeneous phase formation at elevated temperatures and pressures [47]. Regarding ethanolysis monitoring, the sole online method was published by Trevisan et al. [44], who introduced FTIR method for determining fatty acid ethyl esters (FAEE) yield, providing an alternative to 1H-NMR. Nevertheless, using NMR as a reference method for calibration compromises the accuracy due to the lower sensitivity of NMR compared to GC for quantifying minor components [25]. Let it be noted that accuracy is an essential feature of a sensor suitable for reliable control of the ethanolysis process. In summary, online monitoring systems suitable for process control are limited, and the majority of FTIR-based approaches have inherent limitations in accuracy, emphasising the need for further advancements in biodiesel monitoring techniques.

Our previous work focused on a refractive index method for offline monitoring and quality control of FAEE in final biodiesel products [32]. While suitable for batch-end analysis, such offline methods introduce feedback delays and lack real-time control capability. In contrast, this study aims to develop a robust measuring system for the online monitoring of ethanolysis, employing FTIR as a dependable real-time feedback tool for process control. The monitoring process utilised a flow cell on the ATR-FTIR, enabling continuous sampling, while a standardised GC method [20] was concurrently employed for validation. In contrast to common practices of utilising multivariate calibration techniques such as principal component analysis (PCA) [35,36,48,49] or partial least squares (PLS) regression [24,36,40,44] for FTIR calibration models, this study introduces correlation analysis intending to construct a straightforward and reliable linear regression calibration model with accuracy comparable to the GC analysis. The extensive calibration was rigorously validated, demonstrating promising results compared to previously published outcomes employing different approaches. To the best of our knowledge, this is one of the first studies to demonstrate that a simple linear model, trained on real process data and applied to spectral derivatives, can rival PLS in real-time biodiesel production monitoring accuracy.

## 2. Results and Discussion

### 2.1. Calibration Data Acquisition

As detailed in Section 3, the ethanolysis reaction was performed multiple times under varying conditions to ensure comprehensive data collection. Figure 1a shows the progress of the ethanolysis under the selected reaction conditions expressed by the concentration of individual reaction components. The presented data were obtained using a reference GC method [20].

To monitor the progress of ethanolysis and confirm the endpoint of the reaction, the transesterification course was expressed by FAEE content (2) based on offline data from GC analyses. The obtained GC data were subsequently interpolated using an nth order reaction kinetics model, as illustrated in Figure 1b. This allowed the estimation of FAEE content between the sampled points, ensuring that the number of FAEE content values matched the number of spectra obtained from the FTIR measurements.

Figure 2 presents a 3D graph depicting the spectra measured over time during a single experiment. However, extracting meaningful insights from this graph alone is challenging due to the similarities in the spectra of oil and biodiesel. Additionally, the high chemical similarity between TG and FAEE further complicates their differentiation based solely on FTIR spectra. Prior studies have shown visible differences in spectra, but they typically relied on blends, a mixture of pure vegetable oil and final biodiesel, as a calibration dataset. [33,34,40] This work employs real reaction mixtures containing ethanol and catalyst to train the models. Notably, the differences in the spectra of these authentic reaction mixtures are not readily apparent. Using these reaction mixtures, the models can better capture the complexities and variations inherent in the chemical processes, leading to more accurate differentiation between the reaction system states.

### 2.2. Correlation Analysis of the Dataset

Correlation analysis was conducted to identify the relationship between the state of the reaction system determined by GC analysis and FTIR online spectra. This analysis aids in pinpointing regions (i.e., selected wavenumbers) in the spectra where changes in the reaction system are evident. The correlation, documented in Figure 3, is quantified by the coefficient of determination calculated individually for each wavenumber concerning FAEE content. This analysis provides insights into the extent to which variations in spectral features correspond to changes in the reaction system’s state.

After analysing the whole dataset for each experiment individually, it was found a consistent issue occurred at the beginning of some reactions. This problem likely stemmed from the heterogeneity of the reaction mixture at the onset of the reaction. Based on the observation, the empty flow cell equipped on the ATR accessory required more time to establish a stable flow of homogeneous reaction mixture, leading to a delay in obtaining accurate spectra. In this case, measurements commenced only one minute after initiating the pumping of the reaction mixture through the cell. This hypothesis was further supported by evaluating spectra comparisons from a single experiment, as depicted in Appendix A (Appendix A), which clearly shows the spectral instability at reaction initiation and the effect of water vapour on certain spectral regions. Additionally, it was observed that the spectra could be affected by air/vapour (gas bubbles in the reaction mixture) during the flow-through process, particularly evident in the range of approximately 1200–2200 cm^−1^. The described noise slightly increased during the entire reaction; however, in a few reactions, noise was not observed at all. While not explicitly addressed in this work, this observation can serve as a recommendation for further improvements. Following these comprehensive analyses, selected spectra at the beginning of the reaction (obtained before 5 min), which differed from the rest in the set where the cell’s flow-through was not stabilised, were omitted.

Following the discussed removal of outliers—identified by visual inspection as spectra affected by flow instability—approximately 5% of the dataset was excluded. This adjustment resulted in a notable enhancement in correlation, particularly in the spectral regions around 1137, 1206, and 1753 cm^−1^, as depicted in Figure 3.

During the method development, the literature review, and investigation of the gained spectra, it was found that the main peak (1670–1770 cm^−1^) in the spectra of the reaction mixture was moving during the reaction (mainly its leading edge around 1750 cm^−1^). A derivative of the spectra was also used to address this observation. The derivative spectrum enhances peaks and features in the original spectrum, aiding in identifying and analysing individual bands. It helps distinguish closely spaced peaks and resolves overlapping bands not visible in the original spectrum. However, it may amplify noise, so careful management is necessary to prevent compromising data quality [50].

Figure 4 compares the correlation of spectra and spectra derivatives with the FAEE content at different wavenumbers. It is evident that applying the derivative significantly enhanced the correlation in certain regions, particularly around 1750 cm^−1^ and 1000–1300 cm^−1^. Additionally, a new region of improved correlation emerges around 3000 cm^−1^.

Identified regions of significant correlation corroborate existing findings in spectroscopic literature, which frequently underscore specific regions and characteristic peaks associated with ester functional groups. For instance, esters are known to exhibit distinctive peaks in regions such as 1730–1750 cm^−1^ (related to C=O stretching), 1250–1300 cm^−1^ (associated with C-O stretching), and 1050–1150 cm^−1^ (pertaining to C-O-C stretching) [51,52]. This consistency underscores the robustness of the correlation analysis in identifying spectral features that accurately reflect the reaction system’s state.

### 2.3. Development and Comparison of Regression Models

The obtained correlation results prompted the exploration of various approaches. During the development process, the calibration dataset, generated using interpolation (see Section 3.1), was designed. At the same time, a smaller subset derived from GC results served as the validation dataset for the regression model. Validation performance was assessed using only FTIR spectra aligned with actual GC measurements. This ensures an independent comparison with ground-truth reference values and avoids circular validation using interpolated data. The coefficient of determination (R2) and root mean square error on prediction (RMSEP), described in Section 3, were employed as key evaluation metrics for comparing different methods and benchmarking against other approaches in the literature. The PLS method is a common approach found in the literature, usually applied on FTIR data for analysing transesterification reactions [36,40,44]. This method does not require additional correlation analysis, as the PLS algorithm functions independently. More PLS model approaches were performed for comparison to obtain the best possible result, as shown in Figure 5.

Figure 6 illustrates the calibration and validation datasets using the PLS model with 10 components within the range of 650–4000 cm^−1^ of FT-IR spectra. Interestingly, employing more than 10 components did not significantly increase the method’s performance. The coefficient of determination for the validation set reached 0.99, with an RMSEP of 0.94, underscoring the robustness and accuracy of the model within this spectral range.

As was mentioned, the PLS regression is widely used. However, it has its disadvantages. Despite their power, PLS models face challenges with large datasets, complex relationships, and potential overfitting. Outliers and multicollinearity can further impact their performance. Consequently, model interpretation becomes intricate, especially with numerous latent variables (components). Therefore, careful preprocessing and considering alternative methods are essential for reliable outcomes in FTIR data analysis, emphasising thorough model selection, validation, and interpretation.

However, this widely used method was compared with SLR and MLR models. For this purpose, correlation analysis (Section 3.2) was employed to select promising wavenumbers aiming to challenge the PLS method with easy and straightforward linear regression from both obtained raw spectra and spectra derivative. The performance of both tested regression models with a gradual increase in a number of model parameters and thus their complexity is depicted in Figure 5.

As depicted in Figure 5, utilising the PLS method with raw spectra necessitates a minimum of 7 components to achieve similar results as presented in prior studies [40,44] (RMSEP around 2). Conversely, employing the MLR method with only 4 selected wavenumbers—specifically absorbance around 1138, 1206, 1247 and 1753 cm^−1^—yielded a model with an RMSEP value of 1.64. In contrast, applying spectra derivatives enhances precision, requiring fewer components or regions. With spectra derivatives, the PLS model employed in this study needed only 3 components to achieve an RMSEP value below 2. Notably, utilising the MLR model with spectra derivatives yielded even more promising outcomes. Similar results (RMSEP = 2.11) were obtained with only one region (around 1178 cm^−1^), i.e., with simple linear regression.

In prior investigations, Mwenge et al. [40] delved into methanolysis monitoring using an FTIR probe and GC as a reference method, pioneering the development of an FTIR/PLS calibration model that harnessed the power of 11 optimal components. Their efforts culminated in an RMSEP value of 6.32 when spectra were left unprocessed and significantly improved to 2.72 upon application of preprocessing techniques such as mean centring, variance scaling, and spectrum math square root. Similarly, Trevisan et al. [44] presented an online monitoring methodology tailored for transesterification, using ethanol and degummed soybean oil datasets derived from multiple batch reactions, with NMR as the reference method. While NMR is valuable for understanding organic compound structure, it has limitations in quantification—employing NMR as a reference method can compromise accuracy [25]. By employing the versatile PLS method in conjunction with a diverse array of preprocessing methodologies, including derivative, baseline correction, Savitzky–Golay smoothing, mean-centering, auto-scaling, and iPLS, they managed to achieve a commendable RMSEP value of 1.9 under optimal conditions. A detailed comparison of achieved model performance within this study and the discussed literature is available in Table 1.

**Table 1 ijms-26-09381-t001:** Summary of achieved results compared with prior findings.

Regression Method	Data Used	Spectral Range [cm^−1^]	Number of Components/Regions [-]	Reference Method	RMSEP	Source
PLS	preprocessed spectra	650–2500	11	GC	2.72	* Mwenge et al. [40]
PLS	preprocessed spectra	814–3707	6	NMR	1.90	Trevisan et al. [44]
PLS	raw spectra	650–4000	7	GC	1.95	**
PLS	raw spectra	650–4000	10	GC	0.94	** see Figure 6
PLS	spectra derivative	650–4000	3	GC	1.68	**
PLS	spectra derivative	650–4000	7	GC	0.92	**
MLR	raw spectra	650–4000	4	GC	1.64	** see Figure 7
MLR	raw spectra	650–4000	7	GC	1.45	** see Appendix A
SLR	spectra derivative	650–4000	1	GC	2.11	** see Figure 8
MLR	spectra derivative	650–4000	9	GC	1.44	** see Appendix A

* Methanolysis study; ** This study.

Therefore, the results show that applying a novel approach utilising spectral derivatives coupled with simple linear regression achieved comparable RMSEP values as those reported in prior studies [40,44]. Noteworthy, RMSEP value 2.11 was achieved by focusing on just one spectral region, showcasing the efficiency and effectiveness of the introduced method, which is a surprising result of this presented study. Moreover, another impressive finding was gained without resorting to spectra derivative; with only 4 spectral regions in use, it managed to attain a highly competitive model with an RMSEP value of 1.64.

Moreover, the comparison with the well-established PLS method sheds light on the efficacy of the selected approach. While PLS is widely employed in similar analyses, the proposed simple method demonstrated competitive performance. The PLS method, performed in this study for comparison, yielded an RMSEP value of 0.94 using spectra (with 10 components) and 0.92 using spectra derivative (with 7 components).

Overall, the findings underscore the promising prospects of applying spectra derivative and simple linear regression in spectroscopic data analysis, particularly in enhancing accuracy and efficiency while minimising complexity. Refining and optimising these methodologies are aimed at contributing to advancements in analytical techniques and facilitating a more robust and reliable method as a tool for online monitoring systems, not only in biodiesel production.

### 2.4. Method Application

To demonstrate the practical utility of the developed method, the course of the ethanolysis determined by both offline GC analysis and online FTIR measurements in the flow cell was performed as documented in Figure 9. Error bars are included to represent the expected experimental error, even in the case of the reference GC method [20]. Remarkably, the PLS and MLR models exhibited errors lower than 3.5% practically in all gained data points. However, the commonly used PLS model with 10 components demonstrated a similar error rate as the MLR model with only 4 utilised regions. Hence, achieved results confirmed the method practical applicability an online monitoring tool, providing accurate and reliable feedback for process control.

Furthermore, Figure 9 presents the course of 3 different experiments. The red one was initiated at the highest studied temperature and catalyst concentration yielded higher FAEE content—beyond the EU biofuel threshold. Blue and green were initiated under identical conditions; however, one of the experiments resulted in lower FAEE content, likely attributed to a higher water content in the reaction mixture. This led to catalyst consumption in saponification and, consequently, lower FAEE content. This observation underscores the developed method’s ability to discern variations in production process outcomes and simultaneously its capacity to flag failed batches or reaction states that successfully surpass the EU biofuel threshold on the other hand. Therefore, such capabilities enhance the reliability and applicability of the method in real-world scenarios, where control systems must promptly react to changes in the controlled process, even within a wide range of conditions.

While this study focused on refined rapeseed oil, the correlation analysis approach shows promise for adaptation to other feedstocks. The method’s reliance on fundamental ester vibrational modes (C=O stretching at 1730–1750 cm^−1^, C-O stretching at 1250–1300 cm^−1^) suggests applicability across different vegetable oils and even waste oils, though recalibration would be necessary to account for varying impurity levels and fatty acid compositions. The presence of free fatty acids, water, and other contaminants typical in crude or waste oils may require additional spectral preprocessing or alternative correlation regions.

### 2.5. Economic and Industrial Implementation Considerations

The continuous online measurement of reaction progress makes it possible to identify the exact endpoint, so the batch manufacturing process can be stopped as soon as the required FAEE concentration is reached, instead of waiting for a fixed reaction time, which is usual industrial practice. According to the results in this study (see Figure 1 and Figure 9), reaction time can be reduced at the tested conditions by 30 min, i.e., 25% time reduction in the case of a 2 h scheduled reaction. On the other hand, the FTIR system is advantageous in continuous biodiesel plants as well, where automated dosing and process adjustments are essential for consistent product quality. By detecting deviations in real time, the system can trigger corrective actions—such as altering catalyst levels, modifying temperature, or adjusting residence time—without manual intervention.

Integration into plant control infrastructure is also straightforward since modern FTIR spectrometers support standard industrial communication protocols. Depending on the setup, the system can either provide operator notifications or directly drive actuators for closed-loop regulation of catalyst addition, temperature, and reaction completion. Future work will focus on developing advanced process control algorithms that leverage this real-time FAEE content feedback for dynamic process optimisation.

Overall, the proposed FTIR-ATR monitoring system offers significant economic advantages for industrial biodiesel production. Based on current industrial energy prices and market prices for feedstock (ethanol, vegetable oil, and catalyst), economic analysis revealed scale-dependent payback periods. Calculations assumed continuous batch operation (24/7) and an income of 0.05 USD per litre of biodiesel, with payback calculated solely from increased production capacity due to shortened reaction times. For small-scale operations producing approximately 10–15 million litres annually, the initial FTIR equipment investment ($40,000–$60,000) results in payback periods of 3–4 months. For medium-scale plants (100 million litres annually), the payback time reduces to less than a month, demonstrating the scale-dependent economic viability of the technology. Even with smaller time reductions (10%), the payback remains favourable at 1–2 months for medium-scale plants.

## 3. Materials and Methods

### 3.1. Materials

Pure rapeseed oil was obtained from a local grocery store for all experiments. All other used chemicals were of analytical grade.

### 3.2. Small-Scale Transesterification System

A standard industrial-scale manufacturing process was realised at the laboratory level. The different amounts of pure rapeseed oil (500–1000 g) were mixed with ethanol (in a molar ratio of 6:1 to oil) in a glass reactor equipped with a thermometer, reflux condenser, and stirrer. Thereafter, the mixture was heated to the required temperature. After reaching the temperature, a catalyst solution (NaOH in ethanol) was added, marking the initiation of the reaction. The concentration of NaOH varied within the range of 0.25% to 1.00% by weight of the oil. The reaction mixture was vigorously stirred (2000 rpm) for 2 h under reflux at temperatures ranging from 40 °C to 60 °C. Samples were collected at selected time intervals (2, 5, 7, 10, 15, 20, 30, 45, 60, 120 min) from the point of catalyst addition into the reactor. About 2 mL of the reaction mixture were withdrawn and mixed with an n-butanolic solution of adipic acid (0.1 mol/L, 3 mL) in a vial. The mixture was vigorously shaken to halt the reaction promptly. The sample thus obtained was subsequently subjected to GC analysis, following the method described in [20]. A total of 10 ethanolysis experiments were conducted under 5 different condition sets.

### 3.3. Process Monitoring Methods

#### 3.3.1. FTIR

FTIR measurements were performed using an online system with a continuous flow cell, which involved silicon tubing connected to the reactor. A model KNF Lab Simdos 10 membrane pump facilitated the continuous flow. The FTIR analysis was conducted using a Thermo Scientific™ Nicolet™ iS50 FTIR spectrometer (Thermo Fisher Scientific Inc., Madison, WI, USA) with an ATR sampling accessory featuring a diamond crystal. All spectra were collected with a scan number set to 10, a resolution set to 4 cm^−1^, and a background was collected before the experiment—without the cell. The OMNIC^®^ software, provided with the instrument, was used for spectrum collection and initial analysis. The measurement process was automated with a one-minute interval.

#### 3.3.2. GC—Reference Method

Gas chromatography served as the reference method for the analysis of the biodiesel reaction mixture composition, encompassing FAEE, glycerol (*G*), triglycerides (*TG*), diglycerides (*DG*), and monoglycerides (*MG*). The analysis was conducted using a Master GC Fast Gas Chromatograph from DANI Instruments S.p.A., following the procedure outlined in [20]. To ensure the accuracy of the results, each sample underwent at least two injections into the GC chromatograph.

### 3.4. Experimental Data Evaluation

The transesterification reaction involves three successive reverse reactions, converting *TG* (oil/fats) into *DG*, *DG* into *MG*, and *MG* into *G*, as described by (1). An ester molecule (*FAEE*) is produced at each stage of this process, yielding three ester molecules from one TG molecule [8].(1)$$TG+EtOH\overset{catalyst}{↔}DG+FAE\;EDG+EtOH\overset{catalyst}{↔}MG+FAEE\;MG+EtOH\overset{catalyst}{↔}G+FAEE$$

The *FAEE content*, which follows the course of the reaction during the transesterification, was determined using the Formula (2) based on the GC analysis of glycerides and ethyl esters content in the reaction mixture.(2)$$FAEE\;content\;\left[\%\right]\left(\tau \right)=x_{A}=\frac{FAEE\left[\%\right]\left(\tau \right)}{TG\left[\%\right]\left(\tau \right)+DG\left[\%\right]\left(\tau \right)+MG\left[\%\right]\left(\tau \right)+FAEE\left[\%\right]\left(\tau \right)}$$

For each experiment, FTIR spectra were collected continuously at one-minute intervals, yielding 120 spectra per reaction. In parallel, GC analysis was performed at specific times, providing accurate measurements of reaction components. The progress of each reaction, expressed by *FAEE content*, was interpolated using an nth order reaction kinetics model [53]. The kinetic model coefficients were optimised individually for each experiment using Python’s 3.8 curve_fit function from the SciPy v1.13.0 package, employing non-linear least squares to fit the function. This model facilitated interpolation and simulation of the entire reaction course. These interpolated *FAEE* values were aligned with FTIR spectra and used to form the calibration dataset.

To validate model performance, we used the FTIR spectra corresponding only to the time points where GC analyses were actually performed—these served as the validation dataset, as they were associated with measured, not interpolated, *FAEE content*. This strategy ensures that the model is evaluated against true experimental values, not model-generated approximations.

### 3.5. Regression Models and Their Evaluation

Python’s scikit-learn v1.4.1 package was utilised to implement regression models and calculate evaluation metrics.

Python’s LinearRegression function was used to fit linear models—simple linear regression (SLR) and multiple linear regression (MLR). Specifically, the residual sum of squares between the observed targets in the dataset and the targets predicted by the linear approximation was minimised. The PLSRegression function was employed for implementing PLS regression.

The coefficient of determination, often denoted as R-squared (R^2^), and root mean squared error on prediction (RMSEP) were used as key metrics during the models’ development for their evaluation.

## 4. Conclusions

The proposed FTIR monitoring system offers a simple, rapid, precise, and reliable means of online monitoring ethanolysis progress across varied reaction conditions. Specifically, the developed online monitoring system was validated across reaction temperatures ranging from 40 °C to 60 °C and a range of initial catalyst concentrations from 0.25% to 1.0%. The method enables the determination of FAEE content (validated in the range of 40–100%) directly in the reaction mixture without sample pretreatment.

Introducing a novel data evaluation method, this approach employs linear regression, i.e., SLR and MLR, over the more commonly employed PLS regression. Successful utilisation of linear regression necessitated correlation analysis between spectra obtained from multiple reactions and sample composition analysed via the reference GC offline method. Following the removal of outliers, approximately 5% of the dataset, substantial enhancements in correlation were observed, particularly in specific spectral regions of esters (1000–1300 cm^−1^, around 1750 cm^−1^). Further improvements were achieved by applying the spectra derivative.

Both SLR and MLR models exhibited remarkable performance with raw spectra and their derivatives through the identified correlated spectral regions. An RMSEP value of 2.11 was achieved using only one region from the spectra derivative (i.e., in the case of SLR). Additionally, it was shown the method was able to identify discrepancies in the ethanolysis process course effectively and to confirm fulfilling the mandated threshold value prescribed by the European Union standard for biofuel of the batch even during its processing.

These findings underscore the efficacy of FTIR measurements combined with linear regression models as a suitable online monitoring tool for ethyl ester production control and quality assurance. The technology demonstrates strong economic viability with rapid payback periods through increased production capacity, making it attractive even for small-scale biodiesel operations with payback period around 3–4 months. Future work will integrate this monitoring tool into a feedback-controlled reactor system for dynamic biodiesel production optimisation.

## Figures and Tables

**Figure 1 ijms-26-09381-f001:**
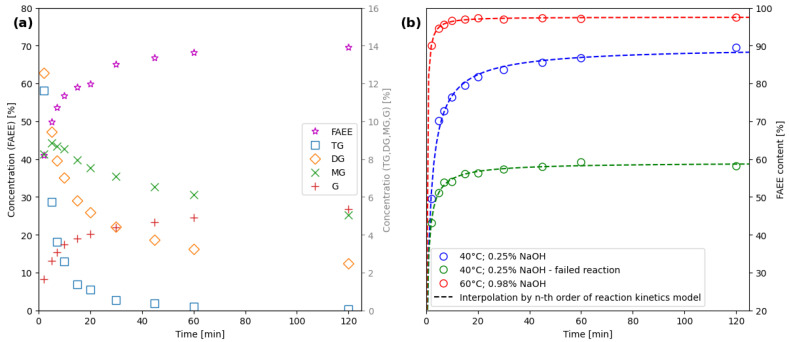
(**a**) GC results from a single experiment (40 °C; 0.25% *w*/*w* NaOH) depict the progression of reaction compounds over time. (**b**) The course of ethanolysis expressed by FAEE content calculated from GC results for each experiment—interpolated by nth order reaction kinetics mode.

**Figure 2 ijms-26-09381-f002:**
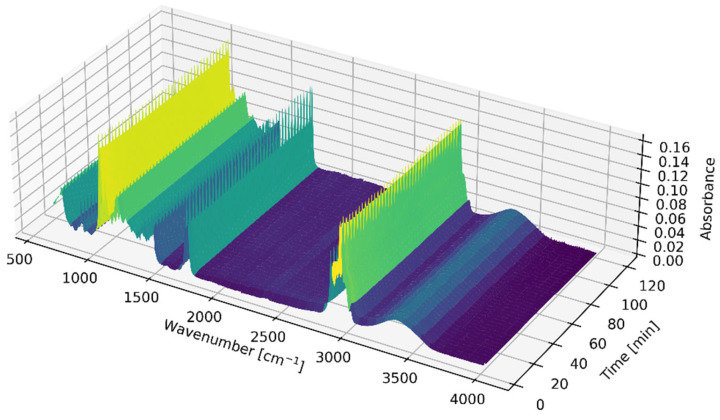
Surface plot—Raw FTIR-ATR Spectra of reaction mixture measured online over time during a single experiment.

**Figure 3 ijms-26-09381-f003:**
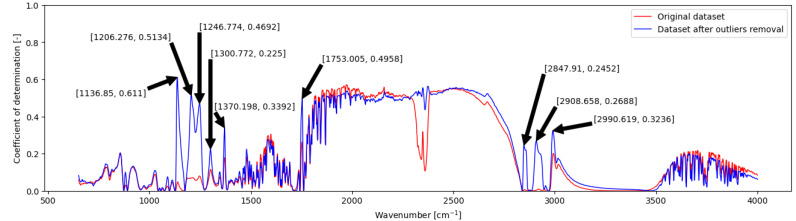
Correlation analysis: Coefficient of determination calculated for each wavenumber concerning FAEE content. Comparison of raw spectra correlation in original dataset vs. dataset without outliers.

**Figure 4 ijms-26-09381-f004:**
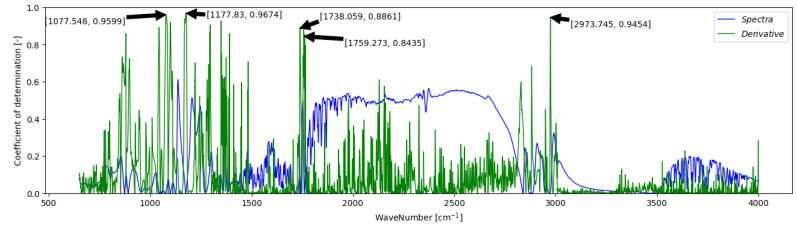
Correlation analysis: Coefficient of determination calculated for each wavenumber concerning FAEE content. Comparison of correlation in raw spectra vs. spectra derivative.

**Figure 5 ijms-26-09381-f005:**
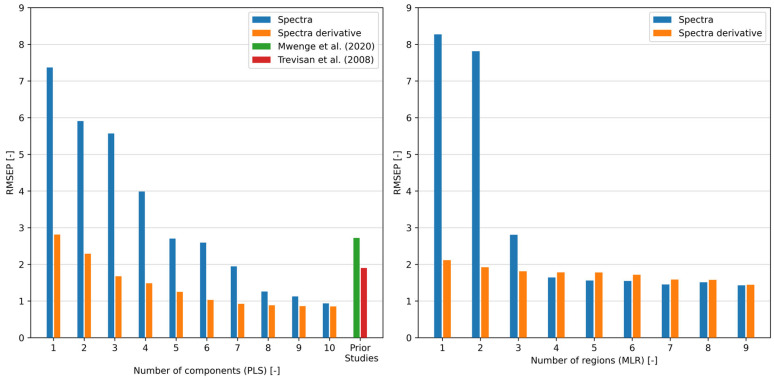
RMSEP dependency on the number of components (PLS)/regions (selected wavenumbers) (MLR) and data (raw spectra vs. spectra derivative), in comparison with prior studies by Mwenge et al. [40] and Trevisan et al. [44].

**Figure 6 ijms-26-09381-f006:**
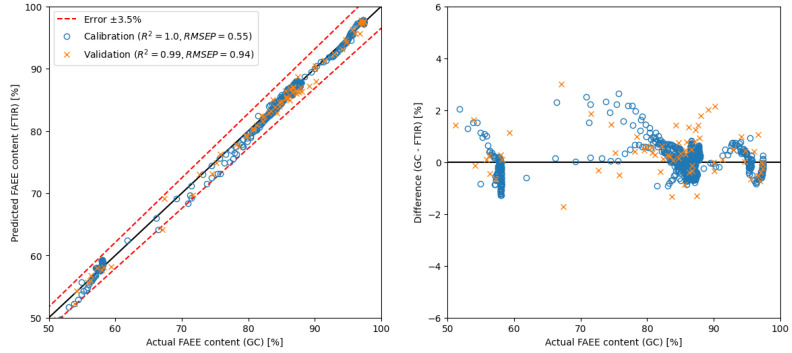
Calibration and validation of the PLS model with 10 components using spectra.

**Figure 7 ijms-26-09381-f007:**
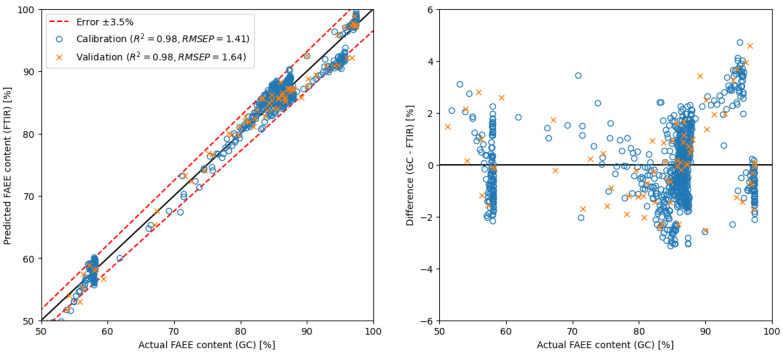
Calibration and validation of MLR model using 4 regions from spectra.

**Figure 8 ijms-26-09381-f008:**
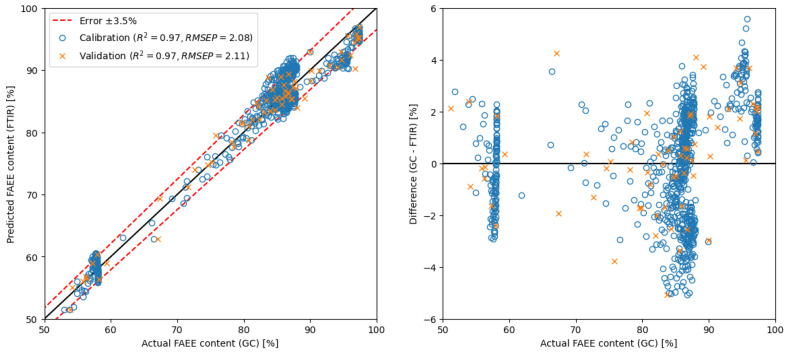
Calibration and validation of SLR model using only 1 region from spectra derivative.

**Figure 9 ijms-26-09381-f009:**
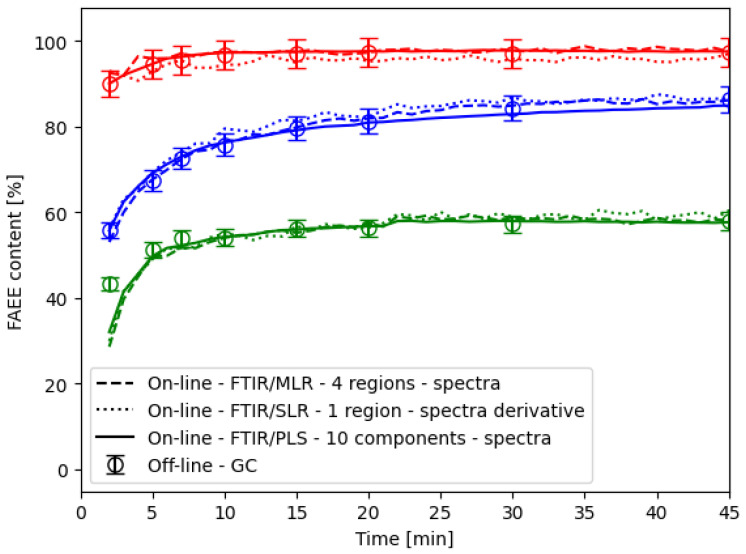
Online FTIR/PLS and FTIR/MLR methods application and comparison with the offline GC analysis. Colors indicate experimental conditions: red (60 °C; 0.98% NaOH), blue (40 °C; 0.25% NaOH), and green (40 °C; 0.25% NaOH; failed reaction), consistent with the legend in Figure 1b.

## Data Availability

The data presented in this study are available upon request from the corresponding author.

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
