# Peer review of "Real-Time FTIR-ATR Spectroscopy for Monitoring Ethanolysis: Spectral Evaluation, Regression Modelling, and Molecular Insight"

_ijms, 2025, doi:10.3390/ijms26199381_

Round 1
Reviewer 1 Report
Comments and Suggestions for Authors
Minor Comments
-
Figure 6 Caption: Clarify what "regions" refer to in the MLR context (i.e., selected wavenumbers). Currently, it may be ambiguous to readers unfamiliar with spectroscopic modeling.
-
Table 1: Consider adding a column for the spectral range used in each study to facilitate comparison.
-
Section 3.4: The use of nth-order kinetics for interpolation is appropriate, but please specify whether the order was optimized per experiment or fixed across all runs.
-
Language Polishing: A few sentences contain minor grammatical issues or awkward phrasing (e.g., line 123: “This study is among the first to demonstrate…” could be strengthened to “To the best of our knowledge, this is one of the first studies…”). A professional language edit is recommended before final publication.
-
Supplementary Material: Ensure that Figure S1 is clearly referenced in the main text (currently mentioned in passing in line 170).
Author Response
Dear Reviewer,
We are grateful for your valuable input. The manuscript was revised and all changes are visible through the track-changes regime. The specific response to your comments follows :
Comment 1: Figure 6 Caption: Clarify what "regions" refer to in the MLR context (i.e., selected wavenumbers). Currently, it may be ambiguous to readers unfamiliar with spectroscopic modelling.
Response 1: Thank you for pointing this out. We clarified in the revised manuscript that “regions” in the context of MLR correspond to selected wavenumbers. This clarification has been added both in Section 2.2 (page 5, line 162) and explicitly in the caption of Figure 6.
Comment 2: Table 1: Consider adding a column for the spectral range used in each study to facilitate comparison.
Response 2: We agree with this comment. Therefore, we have added a new column to Table 1 specifying the spectral range used in each referenced study. This facilitates clearer comparison across different works.
Comment 3: Section 3.4: The use of nth-order kinetics for interpolation is appropriate, but please specify whether the order was optimized per experiment or fixed across all runs.
Response 3: Thank you for the observation. We specified in the revised manuscript that the coefficients of the nth-order kinetics model were optimized individually for each experiment. This clarification can be found in Section 3.4 (page 13, line 418).
Comment 4: Language Polishing: A few sentences contain minor grammatical issues or awkward phrasing (e.g., line 123: “This study is among the first to demonstrate…” could be strengthened to “To the best of our knowledge, this is one of the first studies…”). A professional language edit is recommended before final publication.
Response 4: We appreciate this recommendation. The manuscript has undergone thorough language editing, and several sentences were revised to improve clarity and readability.
Comment 5: Supplementary Material: Ensure that Figure S1 is clearly referenced in the main text (currently mentioned in passing in line 170).
Response 5: We thank the reviewer for this suggestion. We revised the manuscript to clearly reference Supplementary Figure S1. This clarification was added in Section 2.2 (page 6, line 175).

Reviewer 2 Report
Comments and Suggestions for Authors
This study presents the development and validation of an innovative system for the online real-time monitoring of the ethanolysis reaction (for biodiesel production) using FTIR-ATR spectroscopy. The key achievement of the work is the creation of simple yet highly accurate regression models (Simple and Multiple Linear Regression — SLR and MLR), which are comparable in their accuracy to more complex and traditionally used methods, such as PLS. The system was successfully validated across a wide range of industrially relevant conditions (temperature 40–60°C, catalyst concentration 0.25–1.0% w/w) and demonstrated the ability to accurately determine the content of fatty acid ethyl esters (FAEE) in real-time without sample preparation. The prediction error of the developed models was below 3.5% for all data points, which is comparable to the error of the reference gas chromatography method.
Based on the results of the work, there are several clarifying questions:
-
How universal is the proposed method of correlation analysis for identifying significant spectral regions? Will it be equally effective for other types of vegetable oils or other reaction systems, for example, for conducting transesterification reactions in supercritical fluid conditions? There are works on assessing the quantity of the resulting esters using IR spectroscopy; it would be beneficial to add them to the introduction, which would further strengthen the article.
-
What are the estimated payback periods for such an online monitoring system for a typical industrial biodiesel production plant, considering the cost of FTIR equipment and the potential savings from reducing waste and optimizing the process? How is the system proposed to be integrated into existing automated process control systems? Will it only monitor and signal the operator, or is there a possibility for its direct connection to actuators (for example, valves for reagent supply) to create a closed-loop control system?
-
How will the method perform under conditions of high contamination of the reaction mixture or in the presence of impurities characteristic of crude or waste oils?
-
It seems advisable to me to place the results and discussion section after the materials and methods section. This would make the article more comprehensible.
Author Response
Dear Reviewer,
We are grateful for your valuable input. The manuscript was revised and all changes are visible through the track-changes regime. The specific response to your comments follows :
Comment 1: How universal is the proposed method of correlation analysis for identifying significant spectral regions? Will it be equally effective for other types of vegetable oils or other reaction systems, for example, for conducting transesterification reactions in supercritical fluid conditions? There are works on assessing the quantity of the resulting esters using IR spectroscopy; it would be beneficial to add them to the introduction, which would further strengthen the article.
Response 1: Thank you for this important question. We expanded the Introduction to include relevant references to other works employing FTIR for ester quantification (page 3, line 96) and studies applying ATR-FTIR to investigate transesterification mechanisms under supercritical conditions (page 3, lines 98–101). Furthermore, we added a paragraph in Section 2.2 discussing the potential universality of correlation analysis for other vegetable oils and waste oils. While we highlight the promise of adaptability, we also note that recalibration and preprocessing or alternative correlation regions may be necessary due to impurities and contaminants.
Comment 2: What are the estimated payback periods for such an online monitoring system for a typical industrial biodiesel production plant, considering the cost of FTIR equipment and the potential savings from reducing waste and optimising the process? How is the system proposed to be integrated into existing automated process control systems? Will it only monitor and signal the operator, or is there a possibility for its direct connection to actuators (for example, valves for reagent supply) to create a closed-loop control system?
Response 2: We thank the reviewer for raising this important practical aspect. In response, we added a new section titled “Economic and Industrial Implementation Considerations” (Section 3.6, page 12). This section discusses the potential payback period of the FTIR monitoring system in industrial biodiesel production, outlines integration into process control systems, and describes possibilities for both operator signalling and closed-loop control through direct actuator connection. We also briefly highlight these industrial implications in the revised Conclusion (page 14, lines 464-466).
Comment 3: How will the method perform under conditions of high contamination of the reaction mixture or in the presence of impurities characteristic of crude or waste oils?
Response 3: Thank you for this observation. In Section 2.2 (page 11), we added a paragraph discussing the applicability of our method to waste oils. While the method shows promise, we acknowledge that recalibration and potentially different spectral regions or preprocessing methods will be required to handle impurities and contaminants effectively.
Comment 4: It seems advisable to me to place the results and discussion section after the materials and methods section. This would make the article more comprehensible.
Response 4: We appreciate this suggestion. However, the journal specifically requires the order of sections to be Introduction → Results and Discussion → Materials and Methods → Conclusion. Our initially submitted version followed the standard order (Introduction, Materials and Methods, Results and Discussion, Conclusion), but upon editorial request, it was reorganised to the current form. Therefore, the present structure complies with the journal’s requirements. Nevertheless, we leave the final decision on the section order to the Editor.
